# Healthcare Provider Perspectives on Adherence to Adjuvant Endocrine Therapy after Breast Cancer

**Leah K. Lambert** [1,2,*], **Lynda G. Balneaves** [3], **A. Fuchsia Howard** [1], **Stephen L. K. Chia** [2] **and Carolyn C. Gotay** [4]

1   School of Nursing, University of British Columbia, Vancouver, BC V6T 2B5, Canada; fuchsia.howard@ubc.ca
2   BC Cancer, Vancouver, BC V5Z 4E6, Canada; schia@bccancer.bc.ca
3   College of Nursing, Rady Faculty of Health Sciences, University of Manitoba,
    Winnipeg, MB R3T 2N2, Canada; lynda.balneaves@umanitoba.ca
4   School of Population and Public Health, University of British Columbia, Vancouver, BC V6T 1Z3, Canada;
    carolyn.gotay@ubc.ca
*   Correspondence: leah.lambert@ubc.ca

**Abstract:** Adherence to adjuvant endocrine therapy (AET) for breast cancer is suboptimal. The purpose of this study was to: (1) explore the experiences and perspectives of healthcare providers (HCPs) in providing care to breast cancer survivors prescribed AET, (2) identify how social and structural factors influence the provision of AET-related care, and (3) ascertain HCP recommendations for optimizing AET adherence and related care. Individual, in-depth interviews were conducted with 14 HCPs using an interpretive descriptive approach to inquiry and the theoretical lens of relational autonomy. Data was analyzed using thematic and constant comparative techniques. Healthcare providers focused on four main components of AET-related care: (1) the importance of having careful conversations about AET, (2) difficulties in navigating transitions in care, (3) symptom management as a big part of their role, and (4) dealing with AET discontinuation. Recommendations to improve AET adherence focused on developing sustainable and efficient models of delivering high-quality care to women on AET. Healthcare providers play a pivotal role educating women about AET and supporting their adherence to therapy. Sustainable healthcare system innovations and new models of care that address current system gaps are needed to enhance survivorship care, AET adherence, and ultimately, reduce cancer recurrence and mortality.

**Keywords:** adjuvant endocrine therapy; medication adherence; breast cancer; cancer survivorship

## 1. Introduction

Adjuvant endocrine therapy (AET), including tamoxifen and aromatase inhibitors (AI), is part of the standard of care for women with hormone receptor-positive breast cancer. AET reduces the risk of breast cancer recurrence by up to half when taken for at least five years [1]. Despite substantial clinical benefits, a large percentage of women (up to 51%) are non-adherent to AET [2–4], potentially increasing their risk of mortality by 49% [5].

AET non-adherence is complex and influenced by a multitude of factors. The demographic and clinical characteristics (e.g., age, disease severity, comorbidities, toxicity, type of care provider) associated with non-adherence are well documented [6–8]. Psychosocial factors linked to AET adherence include women's perceived necessity for AET, self-efficacy, social support, the quality of the patient-healthcare provider (HCP) relationship, and continuity of follow-up care [8–10]. In addition, qualitative inquiry highlights a lack of understanding of the difficulties women experience adhering to AET [11].

Given the extended time course of AET, the relationship between patients and their HCPs can span several years and multiple providers and healthcare settings. As such, exploring the experiences and perspectives of HCPs caring for women prescribed AET is vital to identifying the broader social and structural factors influencing adherence. The

aims of this study were to: (1) explore HCP experiences and perspectives in providing care to breast cancer survivors prescribed AET; (2) identify how social and structural factors influence the provision of AET-related care; and (3) ascertain HCP recommendations for optimizing AET adherence and related care.

## 2. Methods

This qualitative study employed an interpretive descriptive [12] approach and the theoretical lens of relational autonomy to explore HCP experiences and perspectives related to women's adherence to AET. Relational autonomy focused the inquiry on the broader social and structural context beyond individual factors that influence care decisions related to AET [13]. Patient-reported factors associated with AET adherence [9] and the experiences and perspectives of breast cancer survivors [14] are published elsewhere.

### 2.1. Participant Recruitment

Upon approval from an institutional ethics board, HCPs providing care to women prescribed AET following primary cancer treatment were identified through breast cancer care networks within a Canadian regional cancer agency, including oncologists, general practitioners in oncology, registered nurses, nurse practitioners, and pharmacists. In Canada, general practitioners in oncology are family physicians whose practices are focused on providing care to patients with a diagnosis of cancer. These HCPs provided a range of services related to AET, including prescribing, managing symptoms, and follow-up care. HCPs received a study invitation and consent form via email. Individuals who expressed interest were invited to contact the research team. Informed consent was obtained prior to the interview.

### 2.2. Data Collection and Analysis

Fourteen HCPs were recruited to participate in the study. Semi-structured individual interviews ranging from 16 to 52 min (average 36 min) were conducted (4 in person and 10 by telephone), audio recorded, and transcribed verbatim. An interview guide was developed to explore HCP care experiences and perceptions related to AET, including questions about: (1) how they communicate the benefits and risks of AET, (2) how they provide AET-related follow-up care, (3) why breast cancer survivors experience difficulties with AET adherence, and (4) what strategies would optimize AET adherence. Data was collected and analyzed concurrently, with preliminary data analysis informing questions posed in subsequent interviews [15]. Three members of the research team (LKL, LGB, and AFH) independently reviewed several transcripts to confirm the preliminary coding structure and engaged in an iterative process of discussion throughout analysis. NVivo™ software was used to organize and code data. Thematic analysis [16] was used, first reading transcripts line by line and organizing data into broad codes. Data was then inductively analyzed to conceptualize ideas, comparing the thematic similarities and differences to understand the relationships among data [12]. Memos and conceptual diagrams were developed to advance the analytic process and track categorical and analytic decisions [17].

## 3. Results

The 14 participants included medical oncologists (*n* = 5), radiation oncologists (*n* = 2), nurse practitioners (*n* = 4), a registered nurse, a pharmacist, and a general practitioner in oncology. No family physicians agreed to participate. (see Table 1 for participant characteristics).

HCP commentaries were categorized into four main components of AET-related care: (1) the importance of careful conversations about AET at the start of treatment, (2) difficulties in navigating transitions in care, (3) symptom management as a big part of their role, and (4) dealing with AET discontinuation. Within these components of care, HCPs identified social and structural challenges that influenced AET adherence. It is important to note that medication costs were not a factor influencing adherence in this study; AET is publicly funded in the province of British Columbia for at least five years,

ten years for tamoxifen. Geographic location was also not perceived as a barrier. Rural HCPs described an established system for coordinating the delivery of AET, even in the most remote communities.

**Table 1.** Healthcare Provider Demographic and Practice Characteristics.

| Sample Characteristics *n* = 14 | Frequency (%) |
|---|---|
| **Discipline** | |
| Medical oncologist | 5 (36) |
| Radiation oncologist | 2 (14) |
| General practitioner in oncology | 1 (7) |
| Nurse practitioner | 4 (29) |
| Registered nurse | 1 (7) |
| Pharmacist | 1 (7) |
| **Practice Domain [a]** | |
| Clinical practice | 12 (86) |
| Research | 3 (21) |
| Education | 3 (21) |
| Professional practice | 3 (21) |
| Administration | 4 (29) |
| **Practice Context** | |
| Urban | 6 (42) |
| Rural | 4 (29) |
| Both urban and rural | 4 (29) |
| **Years Worked as a Health Professional** | |
| <20 | 5 (36) |
| >20 | 9 (64) |
| Mean | 19.6 |
| **Years Worked as a Health Professional in Oncology** | |
| <5 | 4 (29) |
| 5–14 | 3 (21) |
| 15–19 | 4 (29) |
| >20 | 3 (21) |
| Mean | 12.5 |

[a] Participants reported more than one domain, resulting in percentages adding to more than 100%.

### 3.1. Careful Conversations

Oncologists in this study were responsible for prescribing and having initial AET treatment discussions with women. Most oncologists reported using a shared decision-making approach in which they personalized information based on a woman's age, risk profile, type of primary cancer treatment(s), and attitudes and beliefs toward AET and medication. They emphasized that it was ultimately a woman's choice to initiate and adhere to AET. Although differences existed in the level of detail shared with women, oncologists framed AET as an efficacious treatment, discussed the benefits and risks, treatment duration and potential for sequential (i.e., switching from tamoxifen to an aromatase inhibitor) and/or extended AET. Some oncologists stressed the importance of accurately portraying AET benefits and risks without being overly persuasive. As a medical oncologist advised, "don't try to oversell it". Furthermore, some HCPs perceived that being realistic about the high likelihood of side effects at the time of AET initiation increased women's long-term tolerance and adherence.

### 3.1.1. Understanding AET Benefits and Risks

Oncologists reported that women's fear of cancer recurrence was not always commensurate with their actual risk, but sometimes exaggerated or underestimated. Consequently, oncologists highlighted their commitment to investing the time to help women understand their actual risk and individual net benefit of AET. Some oncologists described how discussions with low-risk women were more involved because the benefits of AET were less evident than for women with high-risk disease.

*I do have a careful discussion with them about the risks and the benefits. And quite often, it's a case where it's pretty even, so it's not a very clear-cut thing that they should take the medication. . . . sometimes, it's a much longer discussion than women with high-risk disease . . . in low-risk women it's not as clear. (Radiation Oncologist)*

To increase women's understanding of medication benefit, some oncologists used absolute differences when presenting risk statistics due to concerns women might over-estimate the benefits of AET when presented with relative numbers.

*I'm very careful always to give women the absolute numbers, because it really tends to over-estimate the benefit if you just say I'm going to double your chance of not having a recurrence. That can be quite misleading . . . (Radiation Oncologist)*

The protective effect of AET was also described as a challenging concept for some women to embrace.

*There is some thought out there for patients that chemotherapy is the be-all and end-all, and the [AET] piece isn't really that big a deal. But, in fact, [AETs] do reduce their risk of reoccurrence substantially and are really important. (Pharmacist)*

When oncologists took time to explain AET-related concepts (i.e., adjuvant treatment, risk of relapse, microscopic disease), they reported most women accepted the importance of AET. System factors such as time constraints and lack of education resources, however, made these conversations challenging, particularly when women held negative beliefs about medication.

### 3.1.2. Managing Expectations and Concerns about Side Effects

HCPs believed that addressing women's expectations about therapy and ameliorating their fears about potential side effects and longevity of AET upfront positively influenced women to initiate and continue treatment. As shared by one oncologist, potential side effects of AET were extremely concerning for women.

*. . . the general side effects of hot flashes and joint symptoms, mood changes—and then, also getting into the sexual side effects. I think for a lot of women, whether they're young or they're old, that's a big deal. So, we talk about vaginal dryness, vaginal discharge, and loss of libido, and all of those things. Being faced with those side effects . . . that's a problem. (Medical Oncologist)*

When women expressed apprehension about starting AET, oncologists emphasized the importance of addressing women's concerns and encouraging them to try AET, sometimes by framing it as having multiple benefits. For instance, they explained that tamoxifen reduced the risk of recurrence and prevented bone loss among menopausal women. To ease women's fears of serious risks, such as uterine cancer, another oncologist described reassuring women that serious side effects are often detected early because of warning signs such as vaginal bleeding.

When the extended course of AET was a concern for women, an approach that a few oncologists considered effective was to frame the initial decision to start AET as one that women could revisit, emphasizing that their only commitment was to try AET. A medical oncologist offered this advice: "I always say, 'Listen, it's one day at a time and we'll see.'" Another oncologist managed women's hesitancy by suggesting they ease into AET, taking it every other day for the first two weeks, an approach that they found successful among some women.

### 3.2. Navigating Transitions in Follow-Up Care

Generally, oncologists followed practice guidelines recommending patients be discharged to the community following primary cancer treatment [18]. Routinely discharging women on AET, however, was challenging for several reasons: HCP and patient preferences; access to primary care providers; and uncertainty regarding community-based AET-related expertise.

### 3.2.1. Variability in Discharge Practices

There was variability in discharge practices among medical and radiation oncologists and across urban and rural practice contexts. Medical oncologists in urban settings described providing care to women for up to two years of AET and, in some cases, for five years and beyond. Medical oncologists reported following women for a longer period who were at a relatively higher risk of recurrence, enrolled in a clinical trial, without a primary care provider, had lingering side effects or outstanding tests (e.g., mammograms) and/or vocal about their preference to remain under the care of their oncologist. Radiation oncologists, in contrast, typically discharged breast cancer survivors within a few weeks or months after initiating AET to be followed on a longer-term basis by medical oncologists, general practitioners in oncology, or primary care providers.

The discharge practice patterns of oncologists located in rural areas differed due to high patient volumes and relatively fewer oncologists. Rural oncologists shared that they routinely discharged women to the community after one follow-up visit to assess their tolerance of AET.

### 3.2.2. Challenges Transitioning to Primary Care

Some of the HCPs interviewed commented on the substantial number of women that continued to receive survivorship care from oncologists or general practitioners in oncology located in tertiary care settings. This was, in part, due to the difficultly some physicians experienced transitioning women to primary care settings. The perceived shortage of primary care providers and uncertainty regarding their expertise and comfort with AET-related care, may have contributed to these challenges in care transitions.

HCPs who had confidence that a patient's follow-up care would be well managed in the community were more likely to transfer care early in the AET trajectory. A strong relationship between a woman and a knowledgeable primary care provider was believed to result in the same level of adherence as women receiving care from an oncologist. A medical oncologist explained: "If the patient has an excellent relationship with a very good general practitioner, then they'll probably get as good adherence as staying with someone like an oncologist." When transitioning care to the community, oncologists shared a treatment summary and follow-up care plan with primary care providers. The oncologists and the general practitioner in oncology, however, reported many re-referrals from family physicians for symptom management, unnecessary AET prescription refill requests and delays in the start of sequential AET when women were not referred back to their oncologist on time. To mitigate these unsuccessful transitions to primary care, some oncologists and the general practitioners in oncology offered to share follow-up care with family physicians.

> *I'll often say to family doctors, "Listen, I've come to know this patient. This is a patient that will do well in your practice. I'll take care of the heavy lifting. Can you just be the family doctor?" (General Practitioner in Oncology)*

In this shared care model, they were responsible for AET-related care and the family physicians were responsible for women's general healthcare needs.

### 3.3. AET Side Effects: "A Big Part of the Job"

The HCPs described the management of AET-related side effects as a significant challenge in their practice and that disagreement existed over which HCPs ought to be responsible for helping women to manage their symptoms.

### 3.3.1. Responsibility of Symptom Management

Given their specialized knowledge about cancer treatment, medical oncologists became a central point of contact for primary care providers seeking guidance, patient callbacks from the nursing support line and repeat referrals for symptom management. As a result, AET-related side effects consumed a significant portion of their time.

*... Sometimes I'll see them in clinic. But usually, I'll just talk to them on the phone about their endocrine therapy side effects. That probably is the number one task of my week. It's a big part of the job, dealing with hormone therapy. (Medical Oncologist)*

For some physicians, this was problematic as they viewed symptom management among these women as being the responsibility of primary care providers. Other oncologists, however, did not expect primary care providers to have specialized knowledge of AET given their responsibility for a broad range of health issues. A radiation oncologist shared: "I feel that that's my responsibility and I do not mind if they need to come back about their hormone therapy."

### 3.3.2. Symptom Management: Challenges and Solutions

Most HCPs emphasized how side effects were hard to treat due to lack of effective treatment options. A medical oncologist commented: "Other than Venlafaxine, the literature on all the other approaches, alternative and medically tested, there's not much success and most things that have been formally tested have been no better than placebo".

Another challenge to symptom management was the unknown and/or multifactorial etiology of side effects, as well as teasing apart potential confounding factors, particularly when symptoms occurred soon after primary treatment. As a result, HCPs often cautioned women against permanently discontinuing AET before exploring options for improving their symptoms. Some HCPs used medication breaks, stopping AET for a period of time, to evaluate whether a pause improved symptoms. This helped determine the etiology of symptoms and reset women's expectations about therapy, often enabling them to resume AET. Because many women were not keen to take additional medication to treat side effects, HCPs preferred switching AET agents to alleviate symptoms and prevent polypharmacy. As the general practitioner in oncology said: "We don't want to have significant mitigation with several drugs, just to enable the patient to stay on tamoxifen." HCPs shared that some women report routinely skipping doses (e.g., taking AET every other day) to minimize their side effects. However, not all HCPs supported these dose reductions, citing the lack of data for alternative dosing in the adjuvant setting for invasive breast cancer. Other HCPs framed these women's decisions to alter their AET dose from a risk-reduction perspective:

*Once we find out that they may have been doing it every other day and they're finding things to be much more tolerable, we will just support that. Because ultimately, we're hoping to try and get what compliance we can. (Registered Nurse)*

### 3.4. Dealing with AET Discontinuation

Dealing with women's early discontinuation of AET was challenging for HCPs. Not only was AET adherence difficult to monitor across care settings, but women were often reluctant to disclose their decision. HCPs found themselves revisiting initial conversations about AET, and adjusting how risks and benefits were presented based on women's risk status and the amount of time elapsed since stopping AET.

Although HCPs in the study reported there was no standard practice for monitoring AET adherence, all routinely asked about AET use in follow-up visits. Sometimes women sought input and permission from HCPs before deciding to discontinue treatment. In other cases, HCPs were surprised when they learned during follow-up visits that women had already stopped AET. Women who were worried about a recurrence due to ending AET early and had a good relationship with their HCP were perceived as more likely to ask about alternatives or seek permission to discontinue AET. When asked why women discontinue AET without consultation, some HCPs suggested that a culture of compliance may prevent women from voicing their decision to stop AET. It was also suggested women worried about negatively impacting the relationship with their HCP. A nurse practitioner shared what women tell her: "They don't want to upset their oncologists. And a lot of them will come to me and say, 'I saw Doctor So-and-So yesterday. She's not very happy with me because I decided to stop the tamoxifen'" .

When HCPs discovered women had discontinued AET, they were forced to review the case and offer options for resuming treatment, if clinically appropriate. Some women were open to switching agents (e.g., tamoxifen to an AI) or re-starting AET, if HCPs could mitigate bothersome side effects that had led to the decision. For low-risk women, oncologists did not necessarily discourage women from discontinuing AET when they experienced immediate and difficult side effects, describing the absolute benefit of AET for low-risk women as "a very borderline calculation".

> *We're just really talking about doing something reasonable to make that good outlook even better . . . so, if they end up taking tamoxifen and feeling depressed, that's not worth it. If they end up going onto one of the AIs and feeling like they've got bad arthritis all the time and not able to do their usual activities, that's not worth it either. (Radiation Oncologist)*

For women with high-risk disease, HCPs invested more time exploring ways to support AET adherence. Yet, with the passage of time, some women's perception of risk seemed to decrease, and HCPs found it more difficult to convince them of the importance of long-term adherence. A medical oncologist described the dilemma: "Women who have much to gain will almost always persevere with treatment, but not always. And I'm never sure myself, if I should push harder".

### 3.5. Healthcare Provider Recommendations

When queried about how AET adherence could be enhanced, HCPs emphasized the importance of personal connection, patient-HCP relationships, and increased follow-up, particularly in the first three months of AET to assess women's tolerance. Even though a "personal touch", in office or by telephone, was perceived to positively influence women's overall commitment to AET, some HCPs stressed the importance of reducing oncologists' workload related to AET. Oncologists described themselves as an expensive and scarce option for providing long-term AET management, particularly in rural areas.

To improve AET adherence and alleviate demands on oncologists, HCPs emphasized increasing education and support for patients and clinicians, and restructuring the delivery of AET-related care. Participants recommended using other HCPs, such as nurses with specialized knowledge, to deliver AET-related care through survivorship clinics, akin to the services provided by chronic disease clinics (e.g., diabetes). They also identified the need for more comprehensive AET education resources for clinicians and breast cancer survivors, and suggested nurse-led, group education sessions for women prescribed AET. Another suggestion was to develop an online, comprehensive information resource specific to AET that also offered a "symptom checker" and evidence-based solutions to manage side effects.

Other recommendations for supporting AET adherence included: (1) developing visual aids to communicate absolute AET risks and benefits with patients; (2) translating education materials into other languages; (3) creating AET reminder systems; (4) research on non-pharmacological options to alleviate side effects; and (5) upgrading pharmacy systems to generate automated alerts for AET prescription refills.

## 4. Discussion

This study provides a comprehensive portrayal of the unique challenges HCPs face in caring for women on AET. Previous qualitative research with HCPs has mainly focused on investigating a single aspect of AET-related care, such as symptom management [19,20] and physician prescribing patterns [21].

In the current study, HCPs were highly invested in women's AET care and their adherence, but were challenged by social and system factors in four key areas: presenting AET risks and benefits, navigating transitions in care, managing AET-related side effects and dealing with AET discontinuation. In-depth, time consuming and person-centered conversations about AET often occurred to ensure women understood the benefits and risks of AET, while also respecting their values and beliefs. These findings are consistent with previous research that also found women's satisfaction with clinical support [22], the quality

of the patient-physician relationship [23], frequent, patient-centered communication [24,25], preparedness for the possibility of side effects and physician involvement in decision-making [23] were significantly correlated with greater AET adherence.

The HCPs in the current study struggled with the limited consultation time available and a lack of patient education resources. This may, in part, be the result of a growing number of breast cancer survivors requiring long-term, AET-related follow-up care due to increased cancer incidence (a result of an aging population), improved survival rates and longer durations of AET. Consequently, oncologists will be increasingly constrained in providing AET-related care over an extended period of time, making it imperative to coordinate follow-up with primary care providers early in the AET trajectory. The provision of follow-up care was reported to vary widely across HCPs in the study sample and differences were observed among providers and geographical contexts. Similar to our findings, previous research suggests that breast cancer survivors are not always confident their primary care providers have the specialized knowledge necessary to provide comprehensive survivorship care [14,26–28]. Other researchers have also found that primary care providers themselves report inadequate knowledge and training to care for breast cancer survivors [29–31]. Consequently, it will be important to mitigate such barriers in care transitions back to the community for breast cancer survivors on AET.

To overcome challenges associated with providing AET-related care, some HCPs in this study suggested implementing clinics led by nurses to deliver effective, patient-centered care related to AET while also decreasing the burden on primary care and oncology practices. Nurse-led clinics have shown effectiveness in reducing all-cause mortality and major adverse events, as well as improving medication adherence, quality of life and patient satisfaction in chronic disease populations [32–35]. A recent meta-analysis on the effectiveness of medication adherence interventions among patients with coronary artery disease concluded the most effective interventions were delivered by nurses [36]. Currently, there are few nurse-led symptom management programs for women on AET in Canada. If primary care providers, and other HCPs such as nurses, are to assume greater responsibility for long-term management of AET, they will require the knowledge, tools and support to effectively and confidently care for breast cancer survivors.

The negative impact of side effects on women's adherence to AET has been well documented in the literature [6,7,9,37]. HCPs in this study described symptom management as a substantial part of their role. Yet, the lack of symptom management guidelines and the number of patients referred back to oncologists indicate that HCPs may not have access to the professional support required to manage AET-related symptoms in primary care settings. This is consistent with research that found primary care providers expressed uncertainty about their ability to delivery AET-related symptom management and questioned if doing so was beyond their scope of practice [20]. Despite the complexity of symptom management, HCPs need to be aware of the importance of asking about AET symptoms in follow-up consults to ensure women's side effects are not neglected or minimized. Research is also critically needed to identify and evaluate additional strategies for treating AET symptoms for which there are currently no effective management options.

## 5. Limitations

The results of this study are limited by the small sample size, including low representation from several disciplines, and the absence of family physicians' perspectives. The study was conducted in a western Canadian province, which may limit the transferability of findings to other jurisdictions with different models of breast cancer care.

## 6. Conclusions

HCPs play a pivotal role in educating women about AET and supporting their adherence to therapy. Greater support for HCPs, particularly in primary care settings, is needed to address the information and supportive care needs of a growing number of women prescribed AET. Novel ways of addressing the current gaps in care related to breast cancer

survivorship and AET management may need to be developed to enhance survivorship care, AET persistence and ultimately reduce breast cancer risk and mortality.

**Author Contributions:** Conceptualization, L.K.L., L.G.B. and S.L.K.C.; Data curation, L.K.L. and L.G.B.; Formal analysis, L.K.L., L.G.B. and A.F.H.; Funding acquisition, L.K.L.; Investigation, L.K.L., A.F.H., S.L.K.C. and C.C.G.; Methodology, L.K.L., L.G.B., A.F.H., S.L.K.C. and C.C.G.; Project administration, L.K.L. and L.G.B.; Supervision, L.G.B., A.F.H., S.L.K.C. and C.C.G.; Validation, L.K.L., L.G.B., A.F.H., S.L.K.C. and C.C.G.; Writing—original draft, L.K.L.; Writing—review & editing, L.K.L., L.G.B., A.F.H., S.L.K.C. and C.C.G. All authors have read and agreed to the published version of the manuscript.

**Funding:** L.K.L. was supported by a University of British Columbia Four-Year Doctoral Fellowship, a University of British Columbia School of Nursing's Katherine McMillan Director's Discretional Fund research bursary, Registered Nurses Foundation of British Columbia bursaries, and a Canadian Institutes of Health Research Strategic Training in Health Research (CIHR-STIHR) Psychosocial Oncology Research Training doctoral fellowship. Partial support paid by the Canadian Institutes of Health Research-Michael Smith Foundation for Health Research-BC Cancer Health System Impact Postdoctoral Fellowship (H17-167322).

**Institutional Review Board Statement:** The study was conducted according to the guidelines of the Declaration of Helsinki, and approved by the University of British Columbia Behavioral Research Ethics Board (H13-00207, 15 May 2013).

**Informed Consent Statement:** Informed consent was obtained from all subjects involved in the study.

**Data Availability Statement:** The data supporting the study findings are not publicly available, as participants did not give consent for recordings or transcripts to be released to other researchers.

**Acknowledgments:** We thank the healthcare providers who took part in this study for the benefit of women prescribed adjuvant endocrine therapy and improving practice.

**Conflicts of Interest:** The authors declare no conflict of interest. The funders had no role in the design of the study; in the collection, analyses, or interpretation of data; in the writing of the manuscript, or in the decision to publish the results.

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
