# Peer review of "Healthcare Provider Perspectives on Adherence to Adjuvant Endocrine Therapy after Breast Cancer"

_curroncol, doi:10.3390/curroncol28020139_

Round 1
Reviewer 1 Report
Overall summary:
This is a well written, interesting article regarding HCP perspectives on adjuvant endocrine adherence in a Canadian province. The authors highlight a number of issues related to adjuvant endocrine adherence that are relevant every day in clinical practice.
Minor comments:
- On line 179 the authors state “Radiation oncologists, in contrast, typically discharged breast cancer survivors within a few weeks or months after initiating AET due to the low-risk profile of most of their patients”. I am not entirely sure that this statement is accurate. Radiation oncologists typically discharge their patients quickly since they are followed on a long term basis by Medical oncologists, GPO’s, family physicians, not as a result of patients having a low risk profile. Consider revising.
No further comments.
Very well written and concise manuscript.
Reviewer 2 Report
The manuscript by Lambert et. al is well written . I agree with the authors that the major limitation is the sample size. The study would have been more interesting if the sample size was large and more healthcare providers would have agreed to participate. Overall, the manuscript is interesting to read and easy to follow.